# Investigating the Mechanical Performance of Bionic Wings Based on the Flapping Kinematics of Beetle Hindwings

**DOI:** 10.3390/biomimetics9060343

**Published:** 2024-06-06

**Authors:** Chao Liu, Tianyu Shen, Huan Shen, Mingxiang Ling, Guodong Chen, Bo Lu, Feng Chen, Zhenhua Wang

**Affiliations:** 1School of Mechanical and Electrical Engineering, Soochow University, No. 8, Jixue Road, Suzhou 215131, China; c.liu@suda.edu.cn (C.L.); 13382103874@163.com (T.S.); ling_mx@163.com (M.L.); guodongxyz@163.com (G.C.); blu@suda.edu.cn (B.L.); 2College of Mechanical and Electrical Engineering, Nanjing University of Aeronautics and Astronautics, Nanjing 210016, China; shenhuan99@nuaa.edu.cn

**Keywords:** beetle hindwings, flapping kinematics, bionic wing, structural mechanical properties, finite element method

## Abstract

The beetle, of the order Coleoptera, possesses outstanding flight capabilities. After completing flight, they can fold their hindwings under the elytra and swiftly unfold them again when they take off. This sophisticated hindwing structure is a result of biological evolution, showcasing the strong environmental adaptability of this species. The beetle’s hindwings can provide biomimetic inspiration for the design of flapping-wing micro air vehicles (FWMAVs). In this study, the Asian ladybird (*Harmonia axyridis* Pallas) was chosen as the bionic research object. Various kinematic parameters of its flapping flight were analyzed, including the flight characteristics of the hindwings, wing tip motion trajectories, and aerodynamic characteristics. Based on these results, a flapping kinematic model of the Asian ladybird was established. Then, three bionic deployable wing models were designed and their structural mechanical properties were analyzed. The results show that the structure of wing vein bars determined the mechanical properties of the bionic wing. This study can provide a theoretical basis and technical reference for further bionic wing design.

## 1. Introduction

Coleopteran insects’ wings have characteristics not found in other flying insects. Not only are they capable of flying at low Reynolds numbers but their membranous hindwings can also fold. This folding mechanism allows them to adapt when space becomes constrained or flying becomes impractical, enabling their hindwings to automatically tuck beneath the hard elytra. Also, they can swiftly unfold their hindwings again when they take off. Thanks to its deployable hindwings, the beetle has strong environmental adaptability [1], which can provide bionic inspiration for the design of flapping-wing micro air vehicles (FWMAVs).

Before designing the bionic wings of FWMAVs, one of the most important bionic tasks is to study the flapping characteristics of insects, which includes the mechanical properties of their wings, wing kinematics, flight mechanisms, etc. Kinematic study is the most intuitive and visualized method. High-speed photography is often used in the kinematic study of insect flight and flow field visualization experiments [2], which make important contributions to the study of insect flight mechanisms. Early researchers found that the flapping law of different insect populations is essentially a simple harmonic function [3]. Then, others observed the kinematic parameters of insects in hovering and forward flight states and found that the flapping amplitude of insect wings is approximately between 60° and 180° [4], the flapping attack angle is 35° or larger [5], and the flapping frequency is between 25 and 400 Hz [6]. Upon observing the dynamic deformation of butterfly wings, researchers found that the forewings and hindwings of butterflies are independent and can perform flapping movements asymmetrically [7,8]. Experiments have found that damaged fly wings can still compensate for flapping asymmetry, the stroke reversal point, the roll angle [9], and the deviation angle [10] by increasing the flapping amplitude [11], frequency [12], and attack angle [10]. In a study of hovering flight in the damselfly and dragonfly, it was found that the rotation effect and the coupling between the forewing and hindwings will affect the vortex structure and flight performance [13]. In addition, the pattern of a dragonfly’s flapping trajectory also affects its flight aerodynamics [14]. The coupled structure of bees’ wings is composed of a hinge-like structure on the forewing’s trailing edge and the hindwing’s leading edge, which plays an important role in enhancing flight aerodynamics [15]. The shape and vein trend of beetle hindwings, such as those of *Cyrtotrachelus buqueti* [16], will affect wing deformation and aerodynamic performance during flapping [17]. When most insects fly, their translation motion is approximately harmonic [18], and their rotation motion is approximately a trapezoidal function [19].

The mechanical properties of insect wings are influenced by their structural design. Kesel [20] established a physical model based on the forewing of the dragonfly, where the veins were set as solids, and its mechanical load-bearing capacity was analyzed using the finite element method. However, in most insect wings, the veins are hollow tubular structures that can improve the mechanical properties of the wings [21]. For example, the hollow tubular veins of the dragonfly wing structure can prevent stiffness and improve torsional deformation performance [22]. This ladybird hindwing vein structure plays a key role in improving toughness and strength [23,24]. A beetle hindwing model can increase thrust by about 10% compared to a flat wing model [25], and the shape of the beetle’s corrugated hindwings can improve structural strength. Based on the abundant data on insect wings, many researchers have made artificial bionic wings that can mimic the structural features of biological wings. The authors of [26] made an artificial cicada wing with bionic characteristics based on geometric data, including vein thickness, wing mass, and bending stiffness. In [27], the authors developed an artificial wing whose mass, shape, and inherent frequency were similar to those of a hawkmoth wing. They compared the deformation of artificial wings and real wings under high flapping conditions and, although they observed some similarities between the two models, the deformation of the artificial wing was significantly greater. However, there are currently few studies on the mechanical properties of deployable bionic beetle wings. The distribution of bionic wing vein structures also has a significant impact on the load-bearing capacity of bionic wings. Thus, research into the design optimization of bionic wings, based on both kinematic and mechanical properties, can pave the way for developing wings for FWMAVs.

In this study, several kinematic parameters of the flapping flight of the Asian ladybird’s hindwings were studied, including the flight characteristic parameters, the motion trajectory of the wing tips, and the aerodynamic characteristics. On this basis, a model of Asian ladybird flapping kinematics was established through an Euler angle fitting curve. Then, three bionic deployable wing models were designed using Ansys Workbench 2023R1 software, and a structural static simulation was conducted. This research provides a reference for the design of a bionic wing structure for FWMAVs.

## 2. Materials and Methods

### 2.1. Specimens

*H. axyridis* belongs to the order Coleoptera and the family Coccinellidae (Figure 1A). It can effectively prey on various aphids, Lepidoptera larvae, and other pests, and is a common predatory insect in Asia. Its body shape is semi-spherical and it has a wide distribution range. This insect can be seen in the spring and autumn, is easy to catch, and will fly when it needs to feed or mate. The difference between males and females is relatively small. When it is not flying, its hindwings will fold under the elytra. The *H. axyridis* used in this experiment were collected from Suzhou city, Jiangsu Province, China.

### 2.2. Feature Parameters

Five *H. axyridis* were selected, and their weight; body length; body width; hindwing length, width, and area; and folding ratio were measured. The results are shown in Table 1.

### 2.3. Dynamic Capture

The flapping frequency of *H. axyridis* is notably high, characterized by swift movements. To capture and analyze their rapid hindwing motions, a high-speed camera system (Phantom V711, Vision Research Inc., Wayne, NJ, USA) was employed, as depicted in Figure 1B. This system was equipped with a 14-bit CMOS sensor and offered a full-screen resolution of 1280 × 800 pixels. Moreover, it possessed the ability to record up to 6814 frames per second, sufficient for meeting our experimental requirements.

The high-speed camera was connected to a computer via a data cable, and the RJ45 interface on the computer was connected to the data cable of the high-speed camera. Then, the high-speed camera was powered on. When filming the movement of *H. axyridis*’ hindwings, the lens was parallel to the center area. By adjusting the focal length of the high-speed camera, we were able to clearly observe the movement of *H. axyridis*. Then, the frame rate was set to 1000 frames per second after repeated tests, and the resolution was set to 1024 × 800. In addition, four 150 W light-emitting diode (LED) projector lamps were installed to illuminate the experimental area and improve the quality of the image background, as shown in Figure 1C. The room temperature was kept at 28–30 °C to increase the probability of the beetles performing flapping movements [28].

Before video recording, blue acrylic paint was gently applied to the dorsal side of the thorax of *H. axyridis*; the specific location of the paint is denoted by the three blue dots in Figure 1A. The mass of the paint could be ignored (less than 0.05 mg), and no additional physiological effects of this marking on the behavior of the *H. axyridis* were observed. These dots were used as calibration points to calculate the scale in the high-speed recordings. After testing, it was also found that these dots did not affect the natural flapping movement and flexibility of the hindwings. Then, a red fluorescent powder was sprinkled on the tips of the left and right wings (the red dots in Figure 1A), with the purpose of allowing the wing tips to be captured more easily for subsequent data processing [29]. Each beetle’s abdomen was attached to a vertical, 4 cm long wooden rod (diameter = 1.5 mm) using AB glue (Epoxy Resin Strong Liquid Adhesive, HOU-FC220), and the *H. axyridis* were allowed to voluntarily display flight behavior. Wing flapping was recorded for 30 s, and this was repeated 3 times. The initial flapping cycle of at least the first 5 s was disregarded to ensure that transient and ground effects could be ignored.

Three high-speed cameras were placed perpendicular to each other, with the center of the bracket plane set as the coordinate origin. The coordinate plane of Cam. B was set as xoy and the coordinate planes of Cam. A and Cam. C were yoz. Next, the flapping movements of the *H. axyridis* were measured and recorded using an image capture and analysis system (Phantom PCC 2.6 and Simi Motion 8.0).

### 2.4. Wind Tunnel

To obtain the aerodynamic data of *H. axyridis*’ hindwings, wind tunnel experiments were performed in a low-speed, straight-flow wind tunnel at Jilin University, Changchun, China. The main specifications and parameters of the wind tunnel are shown in Table 2. The beetle was fixed on a bracket, which was connected to a force balance (load cell) after adjustment. An LH-SZ-03 load cell (Shanghai Liheng, Shanghai, China; 0–5 N ± 1 mN) was selected. This load cell had the advantages of small size, high precision, and fast response and was suitable for low-speed wind tunnel experiments on the flapping kinematics of small insects.

## 3. Results

### 3.1. Hindwing Wingbeat Motion

Once the flapping motion of the *H. axyridis* stabilized, the flapping cycle could be divided into four stages: downstroke, supination, upstroke, and protonation. During the downstroke, the deformation direction of the *H. axyridis*’ hindwings was perpendicular to the wing plane, and upwards, the middle of the hindwing was raised, the edge bent, and the bending direction was the same as the flapping direction. The deformation shape of the hindwing was like an umbrella, as shown by the green line in Figure 2(A1). During supination, the hindwings were at the lowest angular position of the downstroke. As shown by the green line in Figure 2(A2), the hindwings of the *H. axyridis* flipped upwards from the wing base to the wing tip at the leading edge, and the closer to the wing tip, the greater the flip angle; this motion was held until the end of supination. In addition, the hindwing deformation was opposite to the flapping direction, and a backflow groove was formed at the leading edge (as denoted by the green arrow in Figure 2(A2)). During the upstroke, the deformation direction of the *H. axyridis*’ hindwings was perpendicular to the wing plane and downwards, but the deformation was not obvious and did not form an umbrella-like shape, as shown in Figure 2(A3). During pronation, the hindwings were at the maximum angular position of the upstroke. For the *H. axyridis*, there was almost no downward flip at the position near the base of the leading edge, and the opposite was true at the wing tip, as shown by the green arrow and line in Figure 2(A4). By observing the flapping motion cycle of the *H. axyridis*, the actions of their hindwings at each stage of flight could be explained by aerodynamics. During the downstroke, due to the large pressure difference between the upper and lower wing surfaces, an umbrella-like deformation was caused, which could effectively hold the flight attitude of the *H. axyridis* during flapping [30]. During supination, the *H. axyridis* could capture lift through rotating circulation and wake [31]. During the upstroke, the umbrella-like deformation was no longer obvious; this may have been related to the reduced resistance, which was beneficial for ensuring a stable transition to pronation [30]. At the end of pronation, the lower surface of the hindwing could capture a certain amount of leading-edge vortex (LEV), which was beneficial for generating instantaneous lift and thrust in the next downstroke.

The relevant parameters of the hindwing kinematics can be seen in Table 3. It can be observed that the downstroke of the *H. axyridis*’ hindwings took more time than the upstroke. The downstroke could provide the lift needed for flight, while the upstroke could provide thrust [32]. The time required to generate lift was greater than the time required to generate thrust, indicating that the *H. axyridis* had sufficient lift to maintain the flight attitude. The duration of supination was longer than that of pronation, which allowed the *H. axyridis* to complete complex movements during the supination process, such as braking and turning [31]. Supination was a turning point; at this moment, the hindwings adjusted from the downstroke to the upstroke. At the same time, the hindwings also faced greater resistance. Figure 2D shows the change in the flapping Euler angle of *H. axyridis* hindwings. The position angle *θ*(*t*) gradually decreased during the downstroke and gradually increased during the upstroke, with the flapping amplitudes of *H. axyridis* being approximately 63.3° and 168.4°, respectively; the attack angle *α*(*t*) increased gradually during the downstroke and decreased gradually during the upstroke, and the *α*(*t*) varied from −55.5° to 56.5°. The rotation angle *φ*(*t*) gradually decreased in the first half of the downstroke and gradually increased in the second half of the upstroke. In the cycle of a flapping motion, the change in *φ*(*t*) could be divided into four steps, while *θ*(*t*) and *α*(*t*) could be divided into two steps. The range of change for *φ*(*t*) for the *H. axyridis* was 0° to 92.4°.

### 3.2. Wing Tip Trajectories

Figure 2B shows the 2D and 3D wing tip trajectories of *H. axyridis*. The wing tip trajectory of the *H. axyridis* was a double “8” shape. The intersection points of these trajectories were located near supination and pronation, as shown in Figure 2(B4).

Previous research has shown that the wing tip trajectory patterns of insect flapping mainly fall into three categories: elliptical, single “8”, and double “8” [33]. For example, the wing tip trajectory of a dragonfly’s flapping motion is a single “8” or double “8”, while that of a cicada is elliptical; most insect wings have a single “8” or double “8” wing tip trajectory [34]. With an equal wing surface area, the double “8” wing tip trajectory pattern during flapping yields the greatest lift [33]. *H. axyridis*, with a low mass, wingspan, and wing area, can achieve adequate lift to sustain its flight posture through this double “8” wing tip trajectory. Additionally, the wing tip trajectories of *H. axyridis*’ left and right hindwings are nearly symmetrical, sharing not only the same motion pattern but also the same flapping plane.

Figure 2C shows displacement curves of *H. axyridis*’ wing tips in the *x*, *y*, and *z* directions. In the x direction (Figure 2(C1)), the trajectory of *H. axyridis*’ hindwings can be seen as an approximate sine curve, with the distance between the highest and lowest points being 5.42 ± 6.12 mm. In the y direction (Figure 2(C2)), the motion displacement was very small at only 1.11 ± 2.13 mm, indicating that the thrust required for its forward flight was relatively small, and most of the aerodynamic force was used to generate lift. In the z direction, the trajectory of *H. axyridis*’ hindwings can be seen as an approximate cosine curve. In Figure 2(C1,C2), the difference between the left and right hindwings is very small.

### 3.3. Aerodynamics

The aerodynamics of the hindwings were tested using a wind tunnel force measurement system. The test included the lift and thrust generated by the *H. axyridis* during the wingbeat cycle, which reflected the wingbeat characteristics of *H. axyridis* based on the relationship between the hindwing motion pattern and aerodynamics. Moreover, comparing the aerodynamic results with kinematic parameters is extremely important for studying the mechanical performance of *H. axyridis*. The change in *H. axyridis*’ aerodynamics force with time, observed in the wind tunnel test, is shown in Figure 2E. It can be seen that positive lift was generated during the flapping process of the hindwing, and the average lift was higher than the weight of the *H. axyridis*. When the lift dropped to the minimum, the thrust rose to the peak. In the cycle of wingbeats, *H. axyridis* overcame the effect of lift reduction by increasing thrust. In addition, the aerodynamic regulation of *H. axyridis* was usually closely related to the changes in wingbeat kinematics, and the change in the position of the motion angle determined the aerodynamic characteristics of the hindwing [35]. In detail, the peak lift was 40.05 mN and the peak thrust was 45.88 mN. In addition, there are some similarities in Figure 2D,E. Figure 2D shows that *θ*(*t*) and *φ*(*t*) decreased gradually in the first half of the downbeat. When the hindwing reached its lowest point during the downbeat, the instantaneous aerodynamic lift reached its peak and then the hindwing motion changed from downbeat to upswing; in the second half of the upswing, *θ*(*t*) and *φ(t)* gradually increased. When the hindwing reached its highest point during the upstroke, the instantaneous aerodynamic thrust reached its peak and then the hindwing motion changed from upswing to downbeat. At this point, the *H. axyridis* had completed a cycle of wingbeat actions.

### 3.4. Establishment of the Flapping Kinematic Model

Since *H. axyridis* lack muscles in their hindwings, the wing base serves as the primary mechanism controlling the wingbeat motion of the hindwing. Therefore, the wingbeat can be regarded as a rotational motion around the wing base. From the Euler angles *α*(*t*), *θ*(*t*), and *φ*(*t*), flapping kinematic data can be obtained at any time. Fourier series combined with different coefficients can be used to describe any arbitrary curve [36,37], so the fitted curves of the three flapping angles can be described via the Fourier series. Figure 2D shows the Euler angles of the hindwings during the beetle’s flapping flight. In this study, the first four Fourier terms were chosen to accurately describe the wingbeat motion of the hindwing. The three flapping angles (the angle of attack *α*(*t*), the position angle *θ*(*t*), and the rotation angle *φ*(*t*)) of the hindwing could be defined as shown below:(1)$$\eta \left(t\right)=\sum_{n=0}^{4}\left[a_{n}\cos\left(nKt\right)+b_{n}\sin\left(nKt\right)\right]$$
(2)$$K=\frac{2\pi fc}{2v_{T}}$$
(3)$$v_{T}=2\theta Rf$$
where *η*(*t*) is a function of the three flapping angles over time, *n* is an integer that varies from 0 to 4, *t* is time, *K* is the reduced frequency, an and bn are the constant coefficients of the Fourier terms, *f* is the flapping frequency, *c* is the average chord length, *v_T_* is the reference speed of the wing tip, *R* is the length of the hindwing, and *θ* is the flapping amplitude. For the wing tip trajectory data of *H. axyridis*, the parameters and their constant coefficients used to describe the wingbeat motion in the Fourier series are shown in Table 4 and Table 5.

Kinematic data are key to improving the accuracy of mathematical models. The accuracy of the fitting results was determined using the coefficients of the Fourier series. In previous studies, the wingbeat motion functions of insects mostly used simple sine or cosine functions, which differ significantly from the real wingbeat characteristics of insects [35,38,39]. The real kinematic data of insect wings can guide the design of bio-inspired FWMAVs, so it is crucial to establish an accurate mathematical model of beetle wingbeat motion.

## 4. Discussion

### 4.1. Design of Bionic Wing

Based on the above experimental data and analysis of *H. axyridis*’ hindwing kinematics, a new bionic wing model (named Model I) was designed through the software Space Claim 2023R1, as shown in Figure 3A. Model I, based on the hindwing of *H. axyridis*, consisted of vein bars and a wing membrane, and the contour curve of the wing membrane perfectly matched the *H. axyridis* hindwing. To enhance simulation efficiency and ensure the accuracy of the simulation results, the vein bars were simplified as follows: (1) the design of the leading-edge vein bar and anal posterior vein (AP vein) bar was determined from the *H. axyridis* hindwing; (2) the rest of the veins were arranged under the wing membrane in a uniform and parallel manner and they were connected to the leading-edge vein bar. Furthermore, based on the deployable characteristics of the *H. axyridis* hindwing, the designed bionic wing model increased the axial folding function of the wing through a linkage mechanism. The design details of the bionic wing model (Figure 3) are as follows: Model I was composed of a wing membrane (marked 1), a wing main frame rod (also named connecting rod, marked 2), a rocker (marked 3), a wing root rod (marked 4), a bottom rod (marked 5), a servo crank (marked 6), an AP vein bar (marked 7), and four pin rods (marked 8–11). Two holes (*h* and *j*), located to the right on the wing main frame rod (2), were hinged on the rocker (3) and wing root rod (4) by pin rods (9 and 11, respectively). The lower hole (*m*) of the rocker (3) was hinged on the bottom rod (5) by a pin rod (8). The bottom of the wing root rod (4) was fixed onto the servo crank (6). The bottom hole (*n*) of the servo crank (6) was hinged on the bottom rod (5) by a pin rod (10), and the servo crank (6) was the driving part of the linkage mechanism. The AP vein bar (7) was glued onto the wing root rod (4). Additionally, the wing membrane (1) was glued to the AP vein bar (7), wing root rod (4), and wing main frame rod (2).

### 4.2. Mechanical Performance Analysis of the Bionic Wing Model

#### 4.2.1. Model and Mesh

Based on the design of Model I, two control group models were set, including Models II and III, both of which redistributed the vein bars (Figure 4B,C), differing from Model I (Figure 4A). The details were as follows: Model II exhibited a vein distribution more closely aligned with the natural ladybird hindwing (details in Figure 4D), with primary veins spanning from the base to the entire surface of the wing. Thus, one end of Model II’s vein bars was centrally installed at the right end of the leading-edge vein bar. Model III represented an advanced iteration based on Model II, specifically engineered to mitigate stress concentration at the wing root, thereby enhancing the design of Model II. In particular, Model III incorporated a quarter-arc bar, positioned strategically between the leading-edge vein bar and the AP vein bar, functioning similarly to a stiffener. The remaining vein bars commenced from the right terminus of the leading-edge vein bar, arranged along its extension and interfacing with the quarter-arc bar.

After the wing membranes of the three models were assembled, Models I, II, and III were imported into the “Geometry module” of ANSYS Workbench 2023R1. After automatic recognition, the material properties [40] were assigned to the wing frame (the elasticity modulus was 1.7 × 10^5^ MPa, the density was 1.6 g/cm^3^, Poisson’s ratio was 0.316, and the finite element type was a rod element) and wing membrane (the elasticity modulus was 1.4 × 10^3^ MPa, the density was 0.94 g/cm^3^, Poisson’s ratio was 0.4, and the finite element type was a thin plate element). The contact points between each part were set to the Bonded contact type. Then, three models were imported into the “Meshing module”. The grid was divided using a tetrahedral method, with a mesh size of 0.05 mm. The computational sizes of the three models were as follows: 585,428 elements and 304,824 nodes (Model I), 564,532 elements and 298,324 nodes (Model II), and 565,775 elements and 298,050 nodes (Model III). Due to the irregular shapes of the models, some bad points appeared during meshing. So, the preliminarily divided mesh needed to be modified. A mesh division diagram of the three models is shown in Figure 5A–C,G–I and a mesh division diagram of the three wing frame structures is shown in Figure 5D–F.

#### 4.2.2. Mechanical Performance Analysis

Through the aerodynamic assessments of the *H. axyridis* hindwing in Section 3.3, the peak aerodynamic forces were obtained. These data served as a pivotal reference point, representing the maximum load encountered during flight. Consequently, the ensuing analysis focuses on evaluating the impact of this maximal load on the mechanical integrity of the models’ structures.

The peak force-to-mass ratio (FMR) was determined as the cohesion value between the ladybird wing and the bionic wing model. The FMR could be calculated from the peak aerodynamic force. The mass of an *H. axyridis* was recorded as 3.01 mg, with a corresponding peak lift of 40.05 mN and a peak thrust of 45.88 mN (these data are shown in Figure 2E). Therefore, for a single hindwing, the peak aerodynamic force was calculated to be 30.45 mN. So, the FMR of *H. axyridis* was 10.12 N/g. On the other hand, the mass of the bionic wing model was approximately 1.21 g. The FMR of the model remained equivalent to that of *H. axyridis*, and the model’s peak aerodynamic force was about 12.14 N. Employing this value, structural mechanical analysis was conducted on the three bionic wing models. The same value’s uniform load was applied to the three bionic wing models, with the direction perpendicular to the wing surface. Then, the freedom of the entire wing base was limited. Through a simulation, the deformation, equivalent stress, and equivalent strain of the three models were obtained, as shown in Figure 6.

As shown in Figure 6A–C, the global deformation trend was almost consistent. Under a uniform load, the wing membrane deformation of the three models was larger than that of the wing vein bars. The deformation location of the wing membrane was between the wing vein bars. The deformation gradually increased from the leading edge to the trailing edge along the direction of the wing vein bars. This was because the elastic modulus of the wing membrane of the three models was smaller than that of the wing vein bars. Moreover, the number of wing vein bars at the trailing edge was less than at the leading edge, and there was not enough of a supporting block. So, the deformation of the trailing edge was at the maximum and the deformation of the wing base was at the minimum. There were four areas with significant deformation in Model I. The largest structural deformation of Model I occurred at the lower end of the inner edge. Compared to Model I, the deformation areas of Models II and III were more concentrated. Specifically, significant structural deformation of Model II occurred at the left and right ends of the trailing edge, and the deformation at the left end of the trailing edge was the largest. The largest structural deformation of Model III occurred at the right end of the trailing edge. The maximum deformation was 0.56 mm for Model I, 0.58 mm for Model II, and 0.54 mm for Model III. The maximum deformation of Model III was 7.04% less than that of Model II. In addition, the maximum deformation was not significantly different between the three models. This indicated that the bionic wing model had better structural rigidity. Compared to Models II and III, the deformation areas in Model I were distributed more uniformly, indicating that the structure of Model I was better suited for bearing large wind loads.

Figure 6D–F shows equivalent stress diagrams of the three models under the uniform load. It can be observed that the wing membrane had less stress. Because the vein bars were obscured by the wing membrane, a stress diagram was produced after hiding the wing membrane and is shown in Figure 6G–I. The maximum stress values of the three models were concentrated on the vein bars. This indicated that the vein bars were the primary load-bearing components and played a crucial role in compression resistance. This conclusion is similar to that of previous research on the mechanical properties of butterfly wings [41]. Moreover, the stress values of the vein bars gradually decreased from the leading edge to the trailing edge. The maximum stress of the three models was located on the left end of the leading-edge vein; for Model I, it was 60.26 MPa; for Model II, it was 73.04 MPa; and for Model III, it was 62.65 MPa. Specifically, the maximum stress of Model I was 17.50% and 3.81% less than that of Models II and III. This meant that the bearing strength of Model I was larger than that of Models II and III. Therefore, the vein bar structure of Model I effectively improved the anti-compression performance of the wings. Compared to Models II and III, the design of Model I was more reasonable.

Figure 6J–L shows equivalent strain diagrams of the three models under the uniform load. It can be observed that the maximum strain locations in Models I and II were largely consistent, with both located at the left end of the trailing edge, and their maximum stress locations were located at the left end of the leading edge. The maximum strain location of Model III was at the right end of the trailing edge, but its maximum stress location was at the intersection of the quarter-arc bar and the leading-edge vein bars. Specifically, the maximum strain of Model I was 0.14 mm, while the maximum strains of Models II and III were both 0.12 mm. The maximum strain of Model I was 15.95% higher than that of Models II and III. These higher stress and strain values indicated that this location was easier to destroy. Thus, the structural strength of the AP vein bar of Models I and II was relatively low. In other words, in the future design of bionic deployable wings, the strength of the AP vein bar should be improved by, for example, replacing the materials with those of higher strength or connecting the AP vein bar with other vein bars at the trailing edge to improve the damage resistance of this location.

In summary, the mechanical performance of Model I was optimal among the three models. Additionally, it can be confirmed that our bionic wing design inspired by beetles’ hindwings is effective.

## 5. Conclusions

In this study, the flapping of Asian ladybird (*H. axyridis*) wings during hovering was studied in detail, including wingbeat motion, wing tip trajectories, and flight aerodynamics, using a high-speed camera capture system. The flapping kinematics of the hindwing were measured and analyzed, and the results indicate that the wing tip trajectory of *H. axyridis* has a double “8” pattern. The aerodynamic test verified that the beetle’s high-lift mechanism can provide enough aerodynamic force. Then, the Euler angle definition method was introduced, and the instantaneous flapping angle function of *H. axyridis* in multiple flapping cycles was accurately described using a fourth-order Fourier series. Based on the results of the flapping kinematics analysis, three bionic deployable wing models (Models I, II, and III) imitating the hindwings of *H. axyridis* were designed by SpaceClaim 2023R1 and, based on the results of the aerodynamic test, the mechanical performance of three models was simulated and analyzed using ANSYS Workbench 2023R1. The results show that under the maximum static load during flight, the stress and deformation distribution areas of Model I were relatively uniform, while the stress distribution areas of Models II and III were more concentrated. Finally, we found that each vein bar of Model I played a load-bearing role.

## Figures and Tables

**Figure 1 biomimetics-09-00343-f001:**
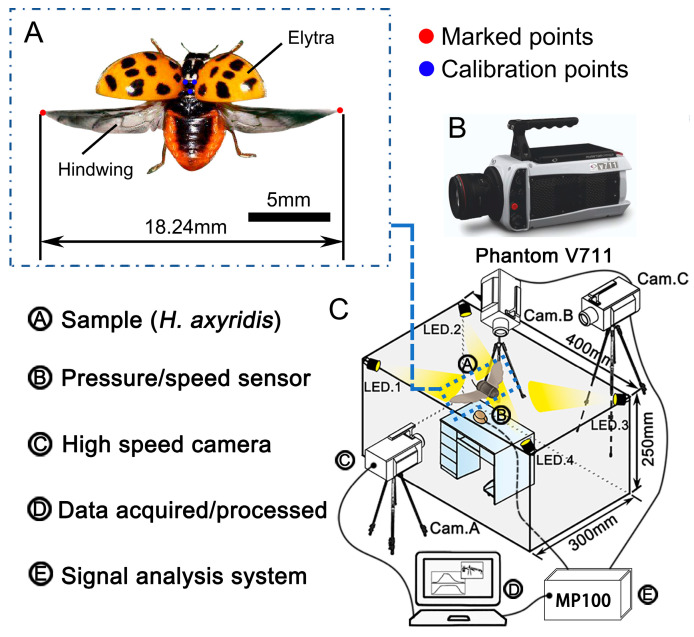
Observation and study of the flapping kinematics of *H. axyridis*’ wing flapping. (**A**) *H. axyridis* beetle specimen. Three blue dots were painted on the dorsal side of the thorax to track the orientation of the body and calculate the scale in high-speed recordings. Red dots were painted on the wing tips to capture them more easily for subsequent data processing. (**B**) The high-speed camera (Phantom V711, Vision Research Inc., Wayne, NJ, USA) with a 14-bit COMS sensor and a full-screen pixel resolution of 1280 × 800 (px). (**C**) Dynamic capture setup for high-speed recording: three synchronized high-speed cameras were placed perpendicular to each other. Four 150 W LED projector lamps were used to illuminate the experimental area and improve the quality of the images.

**Figure 2 biomimetics-09-00343-f002:**
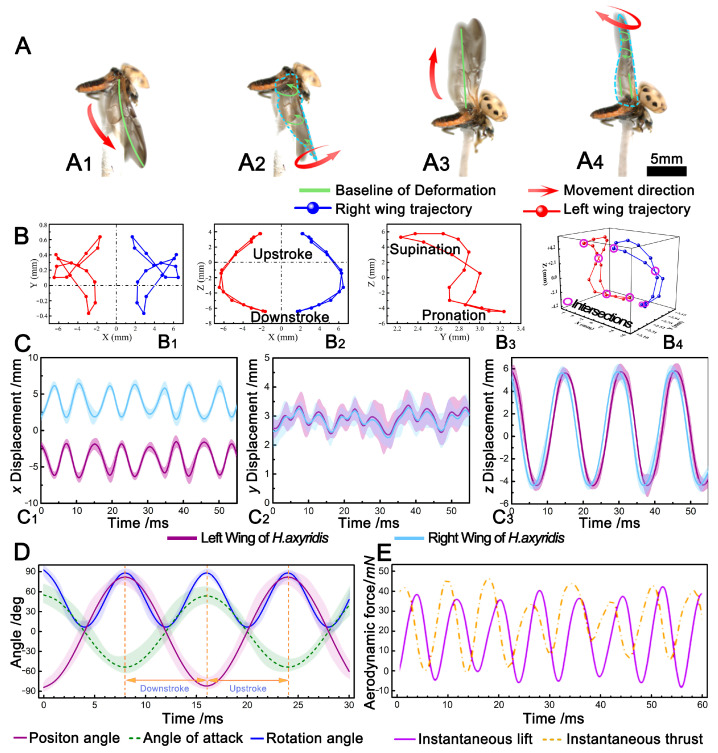
Flight attitude and wing tip trajectory analysis of *H. axyridis*. (**A**) The four stages of the flapping cycle: downstroke, supination, upstroke, and protonation. (**B**) The mean cycle wing tip trajectory of the beetle. From the number of crossing points, the wing tip trajectory followed a double “8” pattern. (**C**) Displacement curve analysis of the wing tip in the *x*, *y*, and *z* directions. (**D**) Curves of the three flapping angles of *H. axyridis* during wingbeat motion. (**E**) Aerodynamic testing of the wingbeat flight cycle of *H. axyridis*.

**Figure 3 biomimetics-09-00343-f003:**
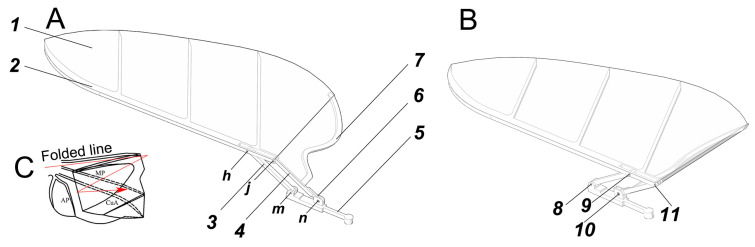
Bionic wing model (Model I) imitating the *H. axyridis* hindwing. Model I was composed of a wing membrane (marked 1), a wing main frame rod (also named connecting rod, marked 2), a rocker (marked 3), a wing root rod (marked 4), a bottom rod (marked 5), a servo crank (marked 6), an AP vein bar (marked 7), and four pin rods (marked 8–11). These holes (*h*, *j*, *m*, and *n*) were the mounting holes of the hinge of the bionic wing model. (**A**) The unfolded state. (**B**) The folded state. (**C**) The folded state of the *H. axyridis* hindwing.

**Figure 4 biomimetics-09-00343-f004:**
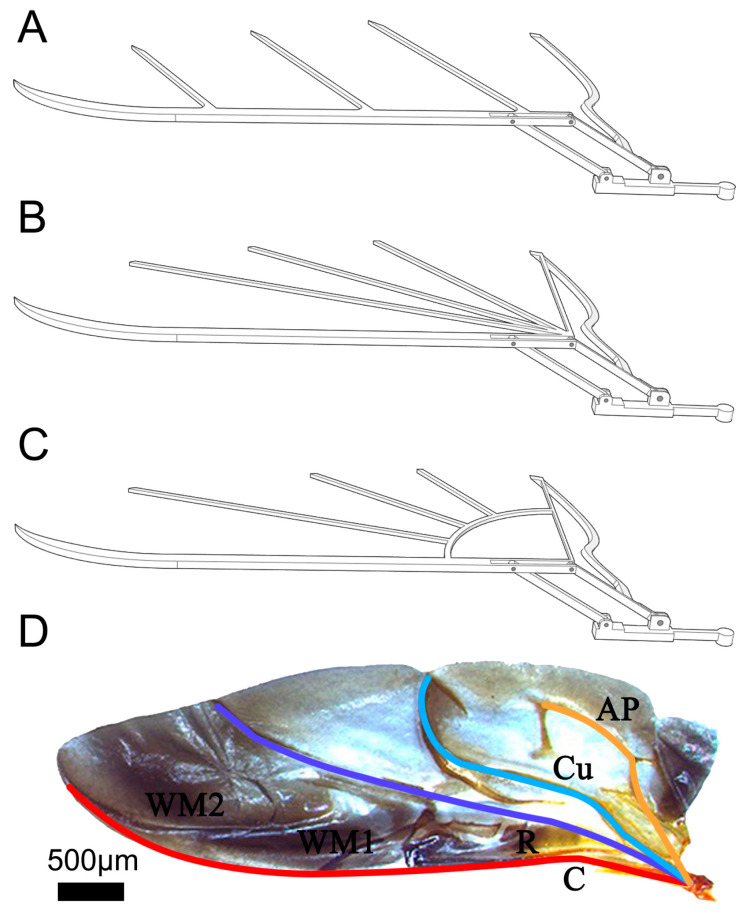
Three bionic wing frame models. (**A**) Frame of Model I. (**B**) Frame of Model II. (**C**) Frame of Model III. (**D**) The natural ladybird hindwing, where C is costa, R is radius, Cu is cubitus, AP is anal posterior, and WM1 and WM2 are the wrinkled membrane.

**Figure 5 biomimetics-09-00343-f005:**
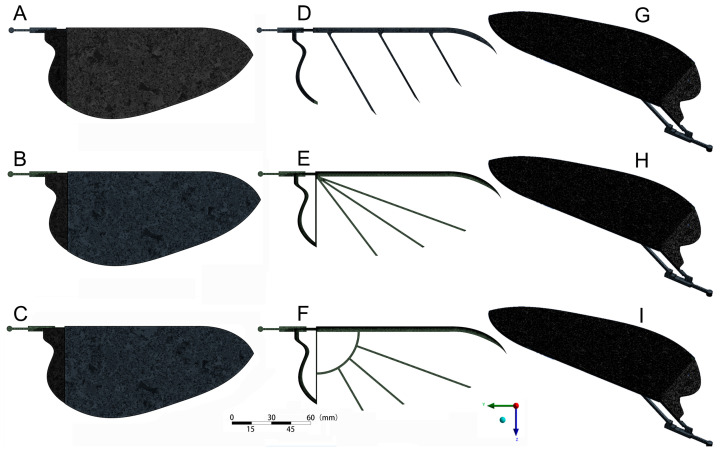
The mesh top view (**A**–**C**) and axonometric diagrams (**G**–**I**) of the three models and three wing frame structures (**D**–**F**).

**Figure 6 biomimetics-09-00343-f006:**
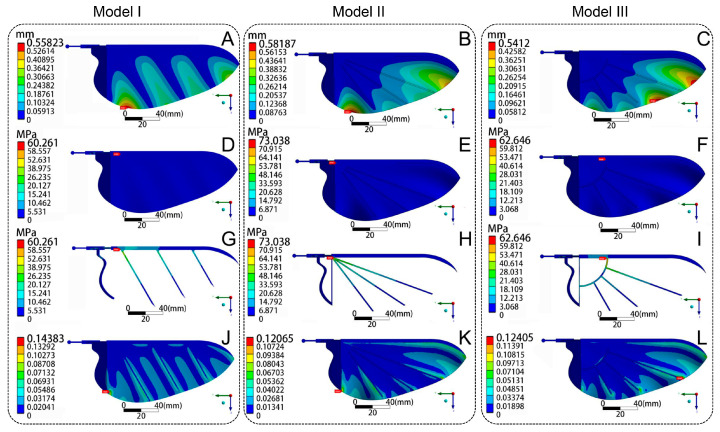
Simulated results of the uniform load test. Total deformation (**A**–**C**), equivalent stress (**D**–**I**), and equivalent strain (**J**–**L**) diagrams for Models I, II, and III, respectively.

**Table 1 biomimetics-09-00343-t001:** The feature parameters of *H. axyridis*.

Feature Parameters	Title 2					Title 3
	1	2	3	4	5	
Mass (mg)	2.95	3.05	3.01	2.88	3.16	3.01 ± 0.09
Body length (mm)	6.05	6.15	7.13	6.82	7.05	6.64 ± 0.45
Body width (mm)	4.78	4.89	4.81	4.92	4.95	4.87 ± 0.06
Hindwing length (mm)	9.35	9.74	9.88	9.82	9.76	9.71 ± 0.19
Hindwing width (mm)	2.89	2.75	2.79	2.73	2.88	2.81 ± 0.07
Hindwing area (mm^2^)	23.56	22.85	23.46	21.73	22.57	22.83 ± 0.66
Folding ratio	2.41	2.25	2.38	2.17	2.23	2.29 ± 0.09

**Table 2 biomimetics-09-00343-t002:** Parameter specifications of the wind tunnel.

Test Section Parameters	Value
Working section shape	Rectangle
Working section area (mm^2^)	650 × 450
Length of working section (mm)	1000
Turbulence intensity (%)	<0.3
Form of wind speed regulator	Hot-wire sensor
Range of wind speed (m/s)	0–10
Airflow nonuniformity of working section (%)	<3

**Table 3 biomimetics-09-00343-t003:** Wing kinematic parameters of *H. axyridis*.

Wingbeat Phase				Stroke Plane Angle (°)	Flapping Amplitude (°)	Flapping Frequency (Hz)
Downstroke	Supination	Upstroke	Pronation			
43.8%	18.6%	31.3%	6.3%	2.1	168.4 ± 3.1	62.5 ± 1.6

**Table 4 biomimetics-09-00343-t004:** Parameters of the Fourier series of the wingbeat motion of *H. axyridis*.

	*K*	*f* (Hz)	*c* (mm)	*v_T_* (m/s)	*R* (mm)	*θ* (°)
*H. axyridis*	0.15	62.5	2.77	3.52	9.71	168.4

**Table 5 biomimetics-09-00343-t005:** Constant coefficients of the Fourier series of *H. axyridis*.

	*a* _0_	*a* _1_	*a* _2_	*a* _3_	*a* _4_	*b* _1_	*b* _2_	*b* _3_	*b* _4_
*α(t)*	−0.83	−34.83	7.17	1.96	2.49	7.25	4.03	3.11	2.59
*θ(t)*	7.49	32.03	−3.31	3.12	3.93	−3.43	0.56	1.24	2.34
*φ(t)*	40.18	4.67	38.12	2.60	3.78	3.57	−8.86	5.67	2.27

## Data Availability

The data presented in this study are available upon request from the corresponding author.

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
