# Peer review of "Investigating the Mechanical Performance of Bionic Wings Based on the Flapping Kinematics of Beetle Hindwings"

_biomimetics, 2024, doi:10.3390/biomimetics9060343_

Round 1

Reviewer 1 Report

Comments and Suggestions for Authors

Reviewer comments

This manuscript demonstrates mechanical performance of bioinspired wing model of the ladybird hindwing. The reviewer renders that the following comments must be addressed and then can be considered for publication after a major revision.

1.     Section Introduction: Why the authors use semicolon, not full stop, to segment each sentence in Line 41-51. This is really awkward. In addition, the reference [7] is not sufficient to support the demonstration on butterfly wing flapping and the authors can refers to and cite the following publications (1)-(2). Meanwhile, publication (3) also can provide more support for the demonstration on insect flight with damaged wing.

(1)   Jantzen, B., Eisner, T. Hindwings are unnecessary for flight but essential for execution of normal evasive flight in Lepidoptera. PNAS 105, 16636–16640 (2008).

(2)   Y. Ma, H.Y. Zhao, T. Ma, J.G. Ning, S. Gorb. Wing coupling mechanism in the butterfly Pieris rapae (Lepidoptera, Pieridae) and its role in taking off. J. Insect Physiol. 131, 104212 (2021).

2.    
The overall structure of the manuscript must be streamlined in a logical sequence. Section 4.1 should be in section Result. Section 4.3.1 should be in Section 2 Materials and Methods.

3.     For FEM work, the authors DID NOT list the basic material parameters of the model, like modulus, density, poisson ratio, etc. This is unacceptable. For the basic materials parameters of insect wings, the authors can refers to and cite the publications as follows:

(3)   N. S. Ha, T. L. Jin, N. S. Goo and H. C. Park, Anisotropy and non-homogeneity of an Allomyrina Dichotoma beetle hind wing membrane, Bioinspiration Biomimetics: 6, 046003 (2011).

4.    
In Discussion section, the authors need give in-depth discussion on the significance of kinematic model on beetle flight based on aerodynamic measurement, as well as why you choose the wing vein pattern of Model II and III, not simple description of FEM result on wing deformation.

5.     From the reviewer side, the kinematic test, aerodynamic test and FEM are independent of each other, without logical connection with each other. For instance, the measured aerodynamic force can be applied to the FEM model? Whether is it possible for the authors to simulate the flapping motion of the FEM model using the measured kinematic motion of the beetles? ... … It is better to demonstrate the correlation of the each part of work, if possible.

6.     Furthermore, the language of the manuscript must corrected and polished, including so many linguistic errors listed below as well as not listed:

Line 43: …; then, …

Line 120: temperature unit

Line 120-123: sentence grammar

Line 181: … a umbrella-like deformation…

Line 230: …. Al-…

Line 279: … where …

Line 284: … to describe the wingbeat motion by Fourier series …

Line 306-307: …designed the bionic wing model increases …

Line 324-325: …which differs from model I…

Line 354: …Through simulate, …???

Line 375: …to disperse of large deformation …???

… …

Comments on the Quality of English Language

The language of the manuscript must corrected and polished.

Reviewer 2 Report

Comments and Suggestions for Authors

Comments on the Quality of English Language

Line 31: the space narrows or unable to flight

The grammar is not correct.

Line 61:  by the of structure design

The grammar is not correct.

Line 82: FMAV

This abbreviation is not defined. 

Round 2

Reviewer 1 Report

Comments and Suggestions for Authors

The manuscript can be accepted for publication in current form.

Comments on the Quality of English Language

The language can be further improved if possible.

Reviewer 2 Report

Comments and Suggestions for Authors

The revised manuscript still lacks sufficient novelty. In addition, the descriptions are still poor. Therefore, the paper should be rejected.

Comment 2:

In experiment-based studies, the reliability of observations is guaranteed by statistics. In the revised manuscript, there is little information from viewpoints of statistics. The information about the number of data, standard deviation, variance, etc. is lacking.

Comment 4:

Whether overfitting occurs also depends on the amount of data. So, there is still possibility of overfitting.

It is needed to draw raw data and fitting curve in the same figure.

Comment 5:

The thickness of the rod elements and that of the plate elements are different. How did you connect these elements?

Besides, the novelty of this work is to consider a foldable wing. Then, the authors proposed the linkage mechanism (Fig 4). However, the linkage mechanism is not considered in the finite element models (Fig 5).

Round 3

Reviewer 2 Report

Comments and Suggestions for Authors

As for comment 5, still I cannot understand how the authors consider the linkage structure.

In reality, linkage structures have some joints, and friction occurs at the joints. In addition, when the wing is exposed to the uniform force, some magnitude of rotation may happen at the joints. 

The manuscript still lacks the information about modeling of the linkage structure.  The more detailed explanations have to be added to the manuscript.
